# Clamping Variables and Approximate Inference

**Adrian Weller**
Columbia University, New York, NY 10027
adrian@cs.columbia.edu

**Tony Jebara**
Columbia University, New York, NY 10027
jebara@cs.columbia.edu

## Abstract

It was recently proved using graph covers (Ruozzi, 2012) that the Bethe partition function is upper bounded by the true partition function for a binary pairwise model that is attractive. Here we provide a new, arguably simpler proof from first principles. We make use of the idea of clamping a variable to a particular value. For an attractive model, we show that summing over the Bethe partition functions for each sub-model obtained after clamping any variable can only raise (and hence improve) the approximation. In fact, we derive a stronger result that may have other useful implications. Repeatedly clamping until we obtain a model with no cycles, where the Bethe approximation is exact, yields the result. We also provide a related lower bound on a broad class of approximate partition functions of general pairwise multi-label models that depends only on the topology. We demonstrate that clamping a few wisely chosen variables can be of practical value by dramatically reducing approximation error.

## 1   Introduction

Marginal inference and estimating the partition function for undirected graphical models, also called Markov random fields (MRFs), are fundamental problems in machine learning. Exact solutions may be obtained via variable elimination or the junction tree method, but unless the treewidth is bounded, this can take exponential time (Pearl, 1988; Lauritzen and Spiegelhalter, 1988; Wainwright and Jordan, 2008). Hence, many approximate methods have been developed.

Of particular note is the Bethe approximation, which is widely used via the *loopy belief propagation* algorithm (LBP). Though this is typically fast and results are often accurate, in general it may converge only to a local optimum of the Bethe free energy, or may not converge at all (McEliece et al., 1998; Murphy et al., 1999). Another drawback is that, until recently, there were no guarantees on whether the returned approximation to the partition function was higher or lower than the true value. Both aspects are in contrast to methods such as the *tree-reweighted* approximation (TRW, Wainwright et al., 2005), which features a convex free energy and is guaranteed to return an upper bound on the true partition function. Nevertheless, empirically, LBP or convergent implementations of the Bethe approximation often outperform other methods (Meshi et al., 2009; Weller et al., 2014).

Using the method of graph covers (Vontobel, 2013), Ruozzi (2012) recently proved that the optimum Bethe partition function provides a lower bound for the true value, i.e. $Z_B \leq Z$, for discrete binary MRFs with submodular log potential cost functions of any arity. Here we provide an alternative proof for attractive binary pairwise models. Our proof does not rely on any methods of loop series (Sudderth et al., 2007) or graph covers, but rather builds on fundamental properties of the derivatives of the Bethe free energy. Our approach applies only to binary models (whereas Ruozzi, 2012 applies to any arity), but we obtain stronger results for this class, from which $Z_B \leq Z$ easily follows. We use the idea of *clamping* a variable and considering the approximate sub-partition functions over the remaining variables, as the clamped variable takes each of its possible values.

Notation and preliminaries are presented in §2. In §3, we derive a lower bound, not just for the standard Bethe partition function, but for a range of approximate partition functions over multi-label

variables that may be defined from a variational perspective as an optimization problem, based only on the topology of the model. In §4, we consider the Bethe approximation for attractive binary pairwise models. We show that clamping any variable and summing the Bethe sub-partition functions over the remaining variables can only increase (hence improve) the approximation. Together with a similar argument to that used in §3, this proves that $Z_B \leq Z$ for this class of model. To derive the result, we analyze how the optimum of the Bethe free energy varies as the singleton marginal of one particular variable is fixed to different values in $[0, 1]$. Remarkably, we show that the negative of this optimum, less the singleton entropy of the variable, is a convex function of the singleton marginal. This may have further interesting implications. We present experiments in §5, demonstrating that clamping even a single variable selected using a simple heuristic can be very beneficial.

## 1.1 Related work

Branching or conditioning on a variable (or set of variables) and approximating over the remaining variables has a fruitful history in algorithms such as branch-and-cut (Padberg and Rinaldi, 1991; Mitchell, 2002), work on resolution versus search (Rish and Dechter, 2000) and various approaches of (Darwiche, 2009, Chapter 8). Cutset conditioning was discussed by Pearl (1988) and refined by Peot and Shachter (1991) as a method to render the remaining topology acyclic in preparation for belief propagation. Eaton and Ghahramani (2009) developed this further, introducing the *conditioned belief propagation* algorithm together with *back-belief-propagation* as a way to help identify which variables to clamp. Liu et al. (2012) discussed feedback message passing for inference in Gaussian (not discrete) models, deriving strong results for the particular class of attractive models. Choi and Darwiche (2008) examined methods to approximate the partition function by deleting edges.

## 2 Preliminaries

We consider a pairwise model with $n$ variables $X_1, \ldots, X_n$ and graph topology $(\mathcal{V}, \mathcal{E})$: $\mathcal{V}$ contains nodes $\{1, \ldots, n\}$ where $i$ corresponds to $X_i$, and $\mathcal{E} \subseteq \mathcal{V} \times \mathcal{V}$ contains an edge for each pairwise relationship. We sometimes consider multi-label models where each variable $X_i$ takes values in $\{0, \ldots, L_i - 1\}$, and sometimes restrict attention to binary models where $X_i \in \mathbb{B} = \{0, 1\} \ \forall i$. Let $x = (x_1, \ldots, x_n)$ be a configuration of all the variables, and $\mathcal{N}(i)$ be the neighbors of $i$. For all analysis of binary models, to be consistent with Welling and Teh (2001) and Weller and Jebara (2013), we assume a reparameterization such that $p(x) = \frac{e^{-E(x)}}{Z}$, where the energy of a configuration, $E = -\sum_{i \in \mathcal{V}} \theta_i x_i - \sum_{(i,j) \in \mathcal{E}} W_{ij} x_i x_j$, with singleton potentials $\theta_i$ and edge weights $W_{ij}$.

### 2.1 Clamping a variable and related definitions

We shall find it useful to examine sub-partition functions obtained by *clamping* one particular variable $X_i$, that is we consider the model on the $n - 1$ variables $X_1, \ldots, X_{i-1}, X_{i+1}, \ldots, X_n$ obtained by setting $X_i$ equal to one of its possible values.

Let $Z|_{X_i=a}$ be the sub-partition function on the model obtained by setting $X_i = a, a \in \{0, \ldots, L_i - 1\}$. Observe that true partition functions and marginals are self-consistent in the following sense:

$$Z = \sum_{j=0}^{L_i-1} Z|_{X_i=j} \ \forall i \in \mathcal{V}, \qquad p(X_i = a) = \frac{Z|_{X_i=a}}{\sum_{j=0}^{L_i-1} Z|_{X_i=j}}. \tag{1}$$

This is not true in general for approximate forms of inference,[1] but if the model has no cycles, then in many cases of interest, (1) does hold, motivating the following definition.

**Definition 1.** We say an approximation to the log-partition function $Z_A$ is *ExactOnTrees* if it may be specified by the variational formula $-\log Z_A = \min_{q \in Q} F_A(q)$ where: (1) $Q$ is some compact space that includes the marginal polytope; (2) $F_A$ is a function of the (pseudo-)distribution $q$ (typically a free energy approximation); and (3) For any model, whenever a subset of variables $\mathcal{V}' \subseteq \mathcal{V}$ is clamped to particular values $P = \{p_i \in \{0, \ldots, L_i - 1\}, \ \forall X_i \in \mathcal{V}'\}$, i.e. $\forall X_i \in \mathcal{V}'$, we constrain

$X_i = p_i$, which we write as $\mathcal{V}' \leftarrow P$, and the remaining induced graph on $\mathcal{V} \setminus \mathcal{V}'$ is acyclic, then the approximation is exact, i.e. $Z_A|_{\mathcal{V}' \leftarrow P} = Z|_{\mathcal{V}' \leftarrow P}$. Similarly, define an approximation to be in the broader class of *NotSmallerOnTrees* if it satisfies all of the above properties except that condition (3) is relaxed to $Z_A|_{\mathcal{V}' \leftarrow P} \geq Z|_{\mathcal{V}' \leftarrow P}$. Note that the Bethe approximation is ExactOnTrees, and approximations such as TRW are NotSmallerOnTrees, in both cases whether using the marginal polytope or any relaxation thereof, such as the cycle or local polytope (Weller et al., 2014).

We shall derive bounds on $Z_A$ with the following idea: Obtain upper or lower bounds on the approximation achieved by clamping and summing over the approximate sub-partition functions; Repeat until an acyclic graph is reached, where the approximation is either exact or bounded. We introduce the following related concept from graph theory.

**Definition 2.** A *feedback vertex set* (FVS) of a graph is a set of vertices whose removal leaves a graph without cycles. Determining if there exists a feedback vertex set of a given size is a classical NP-hard problem (Karp, 1972). There is a significant literature on determining the minimum cardinality of an FVS of a graph $G$, which we write as $\nu(G)$. Further, if vertices are assigned non-negative weights, then a natural problem is to find an FVS with minimum weight, which we write as $\nu_w(G)$. An FVS with a factor 2 approximation to $\nu_w(G)$ may be found in time $O(|\mathcal{V}| + |\mathcal{E}| \log |\mathcal{E}|)$ (Bafna et al., 1999). For pairwise multi-label MRFs, we may create a weighted graph from the topology by assigning each node $i$ a weight of $\log L_i$, and then compute the corresponding $\nu_w(G)$.

## 3 Lower Bound on Approximate Partition Functions

We obtain a lower bound on any approximation that is NotSmallerOnTrees by observing that $Z_A \geq Z_A|_{X_n=j} \; \forall j$ from the definition (the sub-partition functions optimize over a subset).

**Theorem 3.** *If a pairwise MRF has topology with an FVS of size $n$ and corresponding values $L_1, \ldots, L_n$, then for any approximation that is NotSmallerOnTrees, $Z_A \geq \frac{Z}{\prod_{i=1}^{n} L_i}$.*

*Proof.* We proceed by induction on $n$. The base case $n = 0$ holds by the assumption that $Z_A$ is NotSmallerOnTrees. Now assume the result holds for $n-1$ and consider a MRF which requires $n$ vertices to be deleted to become acyclic. Clamp variable $X_n$ at each of its $L_n$ values to create the approximation $Z_A^{(n)} := \sum_{j=0}^{L_n-1} Z_A|_{X_n=j}$. By the definition of NotSmallerOnTrees, $Z_A \geq Z_A|_{X_n=j} \; \forall j$; and by the inductive hypothesis, $Z_A|_{X_n=j} \geq \frac{Z|_{X_n=j}}{\prod_{i=1}^{n-1} L_i}$.

Hence, $L_n Z_A \geq Z_A^{(n)} = \sum_{j=0}^{L_n-1} Z_A|_{X_n=j} \geq \frac{1}{\prod_{i=1}^{n-1} L_i} \sum_{j=0}^{L_n-1} Z|_{X_n=j} = \frac{Z}{\prod_{i=1}^{n-1} L_i}$. $\qquad \square$

By considering an FVS with minimum $\prod_{i=1}^{n} L_i$, Theorem 3 is equivalent to the following result.

**Theorem 4.** *For any approximation that is NotSmallerOnTrees, $Z_A \geq Z e^{-\nu_w}$.*

This bound applies to general multi-label models with any pairwise and singleton potentials (no need for attractive). The bound is trivial for a tree, but already for a binary model with one cycle we obtain that $Z_B \geq Z/2$ for any potentials, even over the marginal polytope. The bound is tight, at least for uniform $L_i = L \; \forall i$.[2] The bound depends only on the vertices that must be deleted to yield a graph with no cycles, not on the number of cycles (which clearly upper bounds $\nu(G)$). For binary models, exact inference takes time $\Theta((|\mathcal{V}| - |\nu(G)|)2^{\nu(G)})$.

## 4 Attractive Binary Pairwise Models

In this Section, we restrict attention to the standard Bethe approximation. We shall use results derived in (Welling and Teh, 2001) and (Weller and Jebara, 2013), and adopt similar notation. The Bethe partition function, $Z_B$, is defined as in Definition 1, where $Q$ is set as the *local polytope* relaxation and $F_A$ is the Bethe free energy, given by $\mathcal{F}(q) = \mathbb{E}_q(E) - S_B(q)$, where $E$ is the energy

and $S_B$ is the Bethe pairwise entropy approximation (see Wainwright and Jordan, 2008 for details). We consider attractive binary pairwise models and apply similar clamping ideas to those used in §3. In §4.1 we show that clamping can never decrease the approximate Bethe partition function, then use this result in §4.2 to prove that $Z_B \leq Z$ for this class of model. In deriving the clamping result of §4.1, in Theorem 7 we show an interesting, stronger result on how the optimum Bethe free energy changes as the singleton marginal $q_i$ is varied over $[0, 1]$.

## 4.1 Clamping a variable can only increase the Bethe partition function

Let $Z_B$ be the Bethe partition function for the original model. Clamp variable $X_i$ and form the new approximation $Z_B^{(i)} = \sum_{j=0}^{1} Z_B|_{X_i=j}$. In this Section, we shall prove the following Theorem.

**Theorem 5.** *For an attractive binary pairwise model and any variable $X_i$, $Z_B^{(i)} \geq Z_B$.*

We first introduce notation and derive preliminary results, which build to Theorem 7, our strongest result, from which Theorem 5 easily follows. Let $q = (q_1, \ldots, q_n)$ be a location in $n$-dimensional pseudomarginal space, i.e. $q_i$ is the singleton pseudomarginal $q(X_i = 1)$ in the local polytope. Let $\mathcal{F}(q)$ be the Bethe free energy computed at $q$ using Bethe optimum pairwise pseudomarginals given by the formula for $q(X_i = 1, X_j = 1) = \xi_{ij}(q_i, q_j, W_{ij})$ in (Welling and Teh, 2001), i.e. for an attractive model, for edge $(i, j)$, $\xi_{ij}$ is the lower root of

$$\alpha_{ij}\xi_{ij}^2 - [1 + \alpha_{ij}(q_i + q_j)]\xi_{ij} + (1 + \alpha_{ij})q_i q_j = 0, \tag{2}$$

where $\alpha_{ij} = e^{W_{ij}} - 1$, and $W_{ij} > 0$ is the strength (associativity) of the log-potential edge weight.

Let $\mathcal{G}(q) = -\mathcal{F}(q)$. Note that $\log Z_B = \max_{q \in [0,1]^n} \mathcal{G}(q)$. For any $x \in [0, 1]$, consider the optimum constrained by holding $q_i = x$ fixed, i.e. let $\log Z_{Bi}(x) = \max_{q \in [0,1]^n : q_i = x} \mathcal{G}(q)$. Let $r^*(x) = (r_1^*(x), \ldots, r_{i-1}^*(x), r_{i+1}^*(x), \ldots, r_n^*(x))$ with corresponding pairwise terms $\{\xi_{ij}^*\}$, be an arg max for where this optimum occurs. Observe that $\log Z_{Bi}(0) = \log Z_B|_{X_i=0}, \log Z_{Bi}(1) = \log Z_B|_{X_i=1}$ and $\log Z_B = \log Z_{Bi}(q_i^*) = \max_{q \in [0,1]^n} \mathcal{G}(q)$, where $q_i^*$ is a location of $X_i$ at which the global optimum is achieved.

To prove Theorem 5, we need a sufficiently good upper bound on $\log Z_{Bi}(q_i^*)$ compared to $\log Z_{Bi}(0)$ and $\log Z_{Bi}(1)$. First we demonstrate what such a bound could be, then prove that this holds. Let $S_i(x) = -x \log x - (1 - x) \log(1 - x)$ be the standard singleton entropy.

**Lemma 6** (Demonstrating what would be a sufficiently good upper bound on $\log Z_B$). *If $\exists x \in [0, 1]$ such that $\log Z_B \leq x \log Z_{Bi}(1) + (1 - x) \log Z_{Bi}(0) + S_i(x)$, then:*
*(i) $Z_{Bi}(0) + Z_{Bi}(1) - Z_B \geq e^m f_c(x)$ where $f_c(x) = 1 + e^c - e^{xc + S_i(x)}$,*
*$m = \min(\log Z_{Bi}(0), \log Z_{Bi}(1))$ and $c = |\log Z_{Bi}(1) - \log Z_{Bi}(0)|$; and*
*(ii) $\forall x \in [0, 1], f_c(x) \geq 0$ with equality iff $x = \sigma(c) = 1/(1 + \exp(-c))$, the sigmoid function.*

*Proof.* (i) This follows easily from the assumption. (ii) This is easily checked by differentiating. It is also given in (Koller and Friedman, 2009, Proposition 11.8). ☐

See Figure 6 in the Supplement for example plots of the function $f_c(x)$. Lemma 6 motivates us to consider if perhaps $\log Z_{Bi}(x)$ might be upper bounded by $x \log Z_{Bi}(1) + (1-x) \log Z_{Bi}(0) + S_i(x)$, i.e. the linear interpolation between $\log Z_{Bi}(0)$ and $\log Z_{Bi}(1)$, plus the singleton entropy term $S_i(x)$. It is easily seen that this would be true if $r^*(q_i)$ were constant. In fact, we shall show that $r^*(q_i)$ varies in a particular way which yields the following, stronger result, which, together with Lemma 6, will prove Theorem 5.

**Theorem 7.** *Let $A_i(q_i) = \log Z_{Bi}(q_i) - S_i(q_i)$. For an attractive binary pairwise model, $A_i(q_i)$ is a convex function.*

*Proof.* We outline the main points of the proof. Observe that $A_i(x) = \max_{q \in [0,1]^n : q_i = x} \mathcal{G}(q) - S_i(x)$, where $\mathcal{G}(q) = -\mathcal{F}(q)$. Note that there may be multiple arg max locations $r^*(x)$. As shown in (Weller and Jebara, 2013), $\mathcal{F}$ is at least thrice differentiable in $(0, 1)^n$ and all stationary points lie in the interior $(0, 1)^n$. Given our conditions, the 'envelope theorem' of (Milgrom, 1999, Theorem

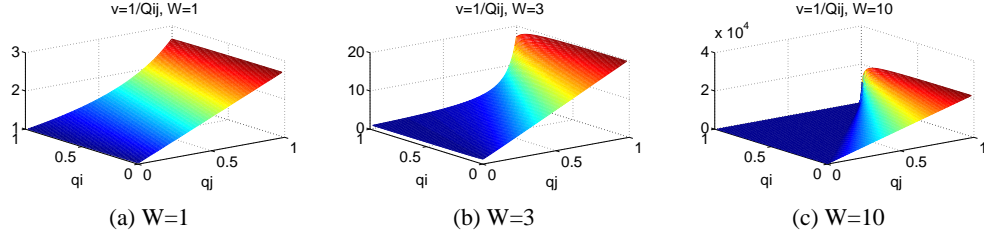

Figure 1: 3d plots of $v_{ij} = Q_{ij}^{-1}$, using $\xi_{ij}(q_i, q_j, W)$ from (Welling and Teh, 2001).

1) applies, showing that $A_i$ is continuous in $[0, 1]$ with right derivative[3]

$$A'_{i+}(x) = \max_{r*(q_i=x)} \frac{\partial}{\partial x} [\mathcal{G}(q_i = x, r^*(x)) - S_i(x)] = \max_{r*(q_i=x)} \frac{\partial}{\partial x} [\mathcal{G}(q_i = x, r^*(x))] - \frac{dS_i(x)}{dx}. \quad (3)$$

We shall show that this is non-decreasing, which is sufficient to show the convexity result of Theorem 7. To evaluate the right hand side of (3), we use the derivative shown by Welling and Teh (2001):

$$\frac{\partial \mathcal{F}}{\partial q_i} = -\theta_i + \log Q_i,$$

where $\log Q_i = \log \dfrac{(1 - q_i)^{d_i - 1}}{q_i^{d_i - 1}} \dfrac{\prod_{j \in \mathcal{N}(i)}(q_i - \xi_{ij})}{\prod_{j \in \mathcal{N}(i)}(1 + \xi_{ij} - q_i - q_j)}$ (as in Weller and Jebara, 2013)

$$= \log \frac{q_i}{1 - q_i} + \log \prod_{j \in \mathcal{N}(i)} Q_{ij}, \text{ here defining } Q_{ij} = \left(\frac{q_i - \xi_{ij}}{1 + \xi_{ij} - q_i - q_j}\right)\left(\frac{1 - q_i}{q_i}\right).$$

A key observation is that the $\log \frac{q_i}{1-q_i}$ term is exactly $-\frac{dS_i(q_i)}{dq_i}$, and thus cancels the $-\frac{dS_i(x)}{dx}$ term at the end of (3). Hence, $A'_{i+}(q_i) = \max_{r*(q_i)} \left[ -\sum_{j \in \mathcal{N}(i)} \log Q_{ij}(q_i, r_j^*, \xi_{ij}^*) \right].$ [4]

It remains to show that this expression is non-decreasing with $q_i$. We shall show something stronger, that at every $\arg\max r^*(q_i)$, and for all $j \in \mathcal{N}(i)$, $-\log Q_{ij}$ is non-decreasing $\Leftrightarrow v_{ij} = Q_{ij}^{-1}$ is non-decreasing. The result then follows since the $\max$ of non-decreasing functions is non-decreasing.

See Figure 1 for example plots of the $v_{ij}$ function, and observe that $v_{ij}$ appears to decrease with $q_i$ (which is unhelpful here) while it increases with $q_j$. Now, in an attractive model, the Bethe free energy is *submodular*, i.e. $\frac{\partial^2 \mathcal{F}}{\partial q_i \partial q_j} \leq 0$ (Weller and Jebara, 2013), hence as $q_i$ increases, $r_j^*(q_i)$ can only increase (Topkis, 1978). For our purpose, we must show that $\frac{dr_j^*}{dq_i}$ is sufficiently large such that $\frac{dv_{ij}}{dq_i} \geq 0$. This forms the remainder of the proof.

At any particular $\arg\max r^*(q_i)$, writing $v = v_{ij}[q_i, r_j^*(q_i), \xi_{ij}^*(q_i, r_j^*(q_i))]$, we have

$$\frac{dv}{dq_i} = \frac{\partial v}{\partial q_i} + \frac{\partial v}{\partial \xi_{ij}} \frac{d\xi_{ij}^*}{dq_i} + \frac{\partial v}{\partial q_j} \frac{dr_j^*}{dq_i}$$

$$= \frac{\partial v}{\partial q_i} + \frac{\partial v}{\partial \xi_{ij}} \frac{\partial \xi_{ij}^*}{\partial q_i} + \frac{dr_j^*}{dq_i}\left( \frac{\partial v}{\partial \xi_{ij}} \frac{\partial \xi_{ij}^*}{\partial q_j} + \frac{\partial v}{\partial q_j} \right). \quad (4)$$

From (Weller and Jebara, 2013), $\frac{\partial \xi_{ij}}{\partial q_i} = \frac{\alpha_{ij}(q_j - \xi_{ij}) + q_j}{1 + \alpha_{ij}(q_i - \xi_{ij} + q_j - \xi_{ij})}$ and similarly, $\frac{\partial \xi_{ij}}{\partial q_j} = \frac{\alpha_{ij}(q_i - \xi_{ij}) + q_i}{1 + \alpha_{ij}(q_j - \xi_{ij} + q_i - \xi_{ij})}$, where $\alpha_{ij} = e^{W_{ij}} - 1$. The other partial derivatives are easily derived: $\frac{\partial v}{\partial q_i} = \frac{q_i(q_j - 1)(1 - q_i) + (1 + \xi_{ij} - q_i - q_j)(q_i - \xi_{ij})}{(1 - q_i)^2(q_i - \xi_{ij})^2}$, $\frac{\partial v}{\partial \xi_{ij}} = \frac{q_i(1 - q_j)}{(1 - q_i)(q_i - \xi_{ij})^2}$, and $\frac{\partial v}{\partial q_j} = \frac{-q_i}{(1 - q_i)(q_i - \xi_{ij})}$.

The only remaining term needed for (4) is $\frac{dr_j^*}{dq_i}$. The following results are proved in the Appendix, subject to a technical requirement that at an $\arg\max$, the reduced Hessian $H_{\backslash i}$, i.e. the matrix of

second partial derivatives of $\mathcal{F}$ after removing the $i$th row and column, must be non-singular in order to have an invertible locally linear function. Call this required property $\mathcal{P}$. By nature, each $H_{\backslash i}$ is positive semi-definite. If needed, a small perturbation argument allows us to assume that no eigenvalue is 0, then in the limit as the perturbation tends to 0, Theorem 7 holds since the limit of convex functions is convex. Let $[n] = \{1, \dots, n\}$ and $G$ be the topology of the MRF.

**Theorem 8.** *For any $k \in [n] \setminus i$, let $C_k$ be the connected component of $G \setminus i$ that contains $X_k$. If $C_k + i$ is a tree, then $\frac{dr_k^*}{dq_i} = \prod_{(s \to t) \in P(i \leadsto k)} \frac{\xi_{st}^* - r_s^* r_t^*}{r_s^*(1 - r_s^*)}$, where $P(i \leadsto k)$ is the unique path from $i$ to $k$ in $C_k + i$, and for notational convenience, define $r_i^* = q_i$. Proof in Appendix (subject to $\mathcal{P}$).*

In fact, this result applies for any combination of attractive and repulsive edges. The result is remarkable, yet also intuitive. In the numerator, $\xi_{st} - q_s q_t = \mathrm{Cov}_q(X_s, X_t)$, increasing with $W_{ij}$ and equal to 0 at $W_{ij} = 0$ (Weller and Jebara, 2013), and in the denominator, $q_s(1 - q_s) = \mathrm{Var}_q(X_s)$, hence the ratio is exactly what is called in finance the beta of $X_t$ with respect to $X_s$.[5]

In particular, Theorem 8 shows that for any $j \in \mathcal{N}(i)$ whose component is a tree, $\frac{dr_j^*}{dq_i} = \frac{\xi_{ij}^* - q_i r_j^*}{q_i(1 - q_i)}$. The next result shows that in an attractive model, additional edges can only reinforce this sensitivity.

**Theorem 9.** *In an attractive model with edge $(i, j)$, $\frac{dr_j^*(q_i)}{dq_i} \geq \frac{\xi_{ij}^* - q_i r_j^*}{q_i(1 - q_i)}$. Proof in Appendix (subject to $\mathcal{P}$).*

Now collecting all terms, substituting into (4), and using (2), after some algebra yields that $\frac{dv}{dq_i} \geq 0$, as required to prove Theorem 7. This now also proves Theorem 5. $\qquad\blacksquare$

### 4.2 The Bethe partition function lower bounds the true partition function

Theorem 5, together with an argument similar to the proof of Theorem 3, easily yields a new proof that $Z_B \leq Z$ for an attractive binary pairwise model.

**Theorem 10** (first proved by Ruozzi, 2012). *For an attractive binary pairwise model, $Z_B \leq Z$.*

*Proof.* We shall use induction on $n$ to show that the following statement holds for all $n$:
If a MRF may be rendered acyclic by deleting $n$ vertices $v_1, \dots, v_n$, then $Z_B \leq Z$.
The base case $n = 0$ holds since the Bethe approximation is ExactOnTrees. Now assume the result holds for $n-1$ and consider a MRF which requires $n$ vertices to be deleted to become acyclic. Clamp variable $X_n$ and consider $Z_B^{(n)} = \sum_{j=0}^1 Z_B|_{X_n = j}$. By Theorem 5, $Z_B \leq Z_B^{(n)}$; and by the inductive hypothesis, $Z_B|_{X_n = j} \leq Z|_{X_n = j} \ \forall j$. Hence, $Z_B \leq \sum_{j=0}^1 Z_B|_{X_n=j} \leq \sum_{j=0}^1 Z|_{X_n=j} = Z$. $\qquad\blacksquare$

## 5 Experiments

For an approximation which is ExactOnTrees, it is natural to try clamping a few variables to remove cycles from the topology. Here we run experiments on binary pairwise models to explore the potential benefit of clamping even just one variable, though the procedure can be repeated. For exact inference, we used the junction tree algorithm. For approximate inference, we used Frank-Wolfe (FW) (Frank and Wolfe, 1956): At each iteration, a tangent hyperplane to the approximate free energy is computed at the current point, then a move is made to the best computed point along the line to the vertex of the local polytope with the optimum score on the hyperplane. This proceeds monotonically, even on a non-convex surface, hence will converge (since it is bounded), though it may be only to a local optimum and runtime is not guaranteed. This method typically produces good solutions in reasonable time compared to other approaches (Belanger et al., 2013; Weller et al., 2014) and allows direct comparison to earlier results (Meshi et al., 2009; Weller et al., 2014). To further facilitate comparison, in this Section we use the same unbiased reparameterization used by Weller et al. (2014), with $E = -\sum_{i \in \mathcal{V}} \theta_i x_i - \sum_{(i,j) \in \mathcal{E}} \frac{W_{ij}}{2} [x_i x_j + (1 - x_i)(1 - x_j)]$.

Test models were constructed as follows: For $n$ variables, singleton potentials were drawn $\theta_i \sim U[-T_{max}, T_{max}]$; edge weights were drawn $W_{ij} \sim U[0, W_{max}]$ for attractive models, or $W_{ij} \sim U[-W_{max}, W_{max}]$ for general models. For models with random edges, we constructed Erdős-Renyi random graphs (rejecting disconnected samples), where each edge has independent probability $p$ of being present. To observe the effect of increasing $n$ while maintaining approximately the same average degree, we examined $n = 10, p = 0.5$ and $n = 50, p = 0.1$. We also examined models on a complete graph topology with 10 variables for comparison with TRW in (Weller et al., 2014). 100 models were generated for each set of parameters with varying $T_{max}$ and $W_{max}$ values.

Results are displayed in Figures 2 to 4 showing average absolute error of $\log Z_B$ vs $\log Z$ and average $\ell_1$ error of singleton marginals. The legend indicates the different methods used: *Original* is FW on the initial model; then various methods were used to select the variable to clamp, before running FW on the 2 resulting submodels and combining those results. *avg Clamp* for $\log Z$ means average over all possible clampings, whereas *all Clamp* for marginals computes each singleton marginal as the estimated $\hat{p}_i = Z_B|_{X_i=1}/(Z_B|_{X_i=0} + Z_B|_{X_i=1})$. *best Clamp* uses the variable which with hindsight gave the best improvement in $\log Z$ estimate, thereby showing the best possible result for $\log Z$. Similarly, *worst Clamp* picks the variable which showed worst performance. Where one variable is clamped, the respective marginals are computed thus: for the clamped variable $X_i$, use $\hat{p}_i$ as before; for all others, take the weighted average over the estimated Bethe pseudomarginals on each sub-model using weights $1 - \hat{p}_i$ and $\hat{p}_i$ for sub-models with $X_i = 0$ and $X_i = 1$ respectively.

maxW and Mpower are heuristics to try to pick a good variable in advance. Ideally, we would like to break heavy cycles, but searching for these is NP-hard. maxW is a simple $O(|\mathcal{E}|)$ method which picks a variable $X_i$ with $\max_{i \in \mathcal{V}} \sum_{j \in \mathcal{N}(i)} |W_{ij}|$, and can be seen to perform well (Liu et al., 2012 proposed the same maxW approach for inference in Gaussian models). One way in which maxW can make a poor selection is to choose a variable at the centre of a large star configuration but far from any cycle. Mpower attempts to avoid this by considering the convergent series of powers of a modified $W$ matrix, but on the examples shown, this did not perform significantly better. See §8.1 in the Appendix for more details on Mpower and further experimental results.

FW provides no runtime guarantee when optimizing over a non-convex surface such as the Bethe free energy, but across all parameters, the average combined runtimes on the two clamped sub-models was the same order of magnitude as that for the original model, see Figure 5.

# 6 Discussion

The results of §4 immediately also apply to any binary pairwise model where a subset of variables may be flipped to yield an attractive model, i.e. where the topology has no frustrated cycle (Weller et al., 2014), and also to any model that may be reduced to an attractive binary pairwise model (Schlesinger and Flach, 2006; Zivny et al., 2009). For this class, together with the lower bound of §3, we have sandwiched the range of $Z_B$ (equivalently, given $Z_B$, we have sandwiched the range of the true partition function $Z$) and bounded its error; further, clamping any variable, solving for optimum $\log Z_B$ on sub-models and summing is guaranteed to be more accurate than solving on the original model. In some cases, it may also be faster; indeed, some algorithms such as LBP may fail on the original model but perform well on clamped sub-models.

Methods presented may prove useful for analyzing general (non-attractive) models, or for other applications. As one example, it is known that the Bethe free energy is convex for a MRF whose topology has at most one cycle (Pakzad and Anantharam, 2002). In analyzing the Hessian of the Bethe free energy, we are able to leverage this to show the following result, which may be useful for optimization (proof in Appendix; this result was conjectured by N. Ruozzi).

**Lemma 11.** *In a binary pairwise MRF (attractive or repulsive edges, any topology), for any subset of variables $S \subseteq \mathcal{V}$ whose induced topology contains at most one cycle, the Bethe free energy (using optimum pairwise marginals) over $S$, holding variables $\mathcal{V} \setminus S$ at fixed singleton marginals, is convex.*

In §5, clamping appears to be very helpful, especially for attractive models with low singleton potentials where results are excellent (overcoming TRW's advantage in this context), but also for general models, particularly with the simple maxW selection heuristic. We can observe some decline in benefit as $n$ grows but this is not surprising when clamping just a single variable. Note, however,

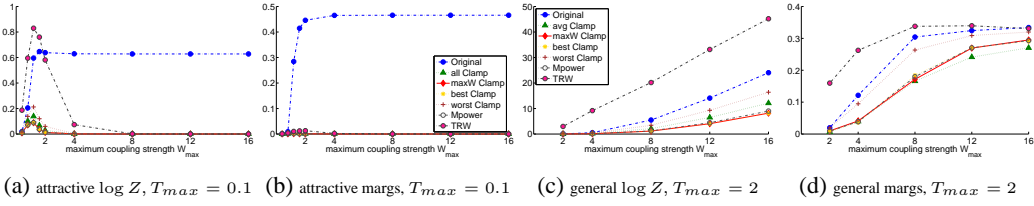

(a) attractive log $Z$, $T_{max} = 0.1$    (b) attractive margs, $T_{max} = 0.1$    (c) general log $Z$, $T_{max} = 2$    (d) general margs, $T_{max} = 2$

Figure 2: Average errors vs true, **complete graph on $n = 10$. TRW in pink**. Consistent legend throughout.

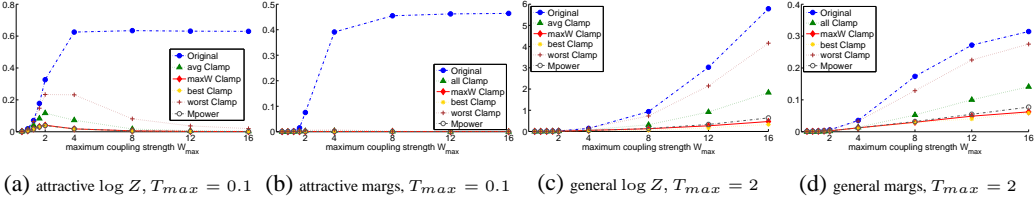

(a) attractive log $Z$, $T_{max} = 0.1$    (b) attractive margs, $T_{max} = 0.1$    (c) general log $Z$, $T_{max} = 2$    (d) general margs, $T_{max} = 2$

Figure 3: Average errors vs true, **random graph on $n = 10, p = 0.5$**. Consistent legend throughout.

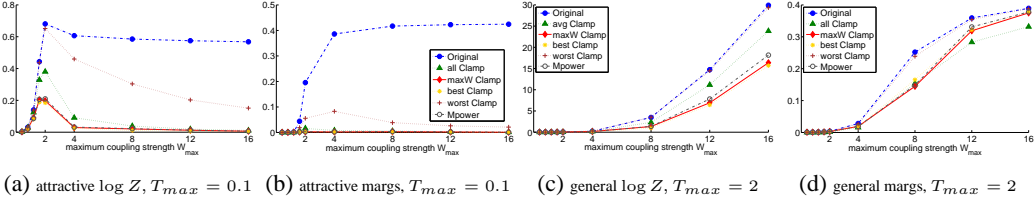

(a) attractive log $Z$, $T_{max} = 0.1$    (b) attractive margs, $T_{max} = 0.1$    (c) general log $Z$, $T_{max} = 2$    (d) general margs, $T_{max} = 2$

Figure 4: Average errors vs true, **random graph on $n = 50, p = 0.1$**. Consistent legend throughout.

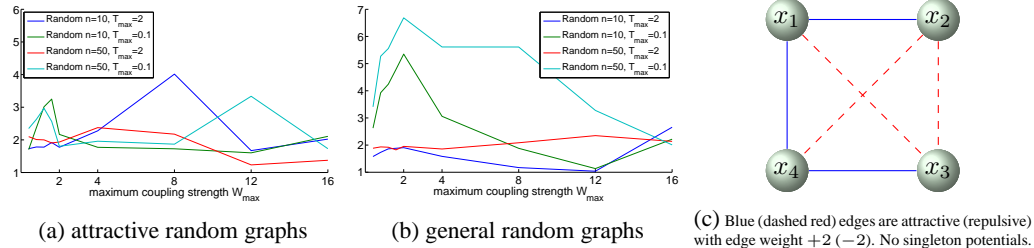

(a) attractive random graphs      (b) general random graphs      (c) Blue (dashed red) edges are attractive (repulsive) with edge weight $+2$ $(-2)$. No singleton potentials.

Figure 5: Left: Average ratio of combined sub-model runtimes to original runtime (using maxW, other choices are similar). Right: Example model where *clamping any variable worsens* the Bethe approximation to $\log Z$.

that non-attractive models exist such that clamping and summing over *any variable* can lead to a *worse* Bethe approximation of $\log Z$, see Figure 5c for a simple example on four variables.

It will be interesting to explore the extent to which our results may be generalized beyond binary pairwise models. Further, it is tempting to speculate that similar results may be found for other approximations. For example, some methods that upper bound the partition function, such as TRW, might always yield a lower (hence better) approximation when a variable is clamped.

**Acknowledgments.** We thank Nicholas Ruozzi for careful reading, and Nicholas, David Sontag, Aryeh Kontorovich and Tomaž Slivnik for helpful discussion and comments. This work was supported in part by NSF grants IIS-1117631 and CCF-1302269.

## Footnotes

[1]For example, consider a single cycle with positive edge weights. This has $Z_B < Z$ (Weller et al., 2014), yet after clamping any variable, each resulting sub-model is a tree hence the Bethe approximation is exact.

[2]For example, in the binary case: consider a sub-MRF on a cycle with no singleton potentials and uniform, very high edge weights. This can be shown to have $Z_B \approx Z/2$ (Weller et al., 2014). Now connect $\nu$ of these together in a chain using very weak edges (this construction is due to N. Ruozzi).

[3]This result is similar to Danskin's theorem (Bertsekas, 1995). Intuitively, for multiple $\arg\max$ locations, each may increase at a different rate, so here we must take the $\max$ of the derivatives over all the $\arg\max$.

[4]We remark that $Q_{ij}$ is the ratio $\left(\frac{p(X_i=1, X_j=0)}{p(X_i=0, X_j=0)}\right) \Big/ \left(\frac{p(X_i=1)}{p(X_i=0)}\right) = \frac{p(X_j=0|X_i=1)}{p(X_j=0|X_i=0)}$.

[5]Sudderth et al. (2007) defined a different, symmetric $\beta_{st} = \frac{\xi_{st} - q_s q_t}{q_s(1 - q_s) q_t(1 - q_t)}$ for analyzing loop series. In our context, we suggest that the ratio defined above may be a better Bethe beta.

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
