[Supplementary Material]

# APPENDIX: SUPPLEMENTARY MATERIAL FOR
## *CLAMPING VARIABLES AND APPROXIMATE INFERENCE*

In this Appendix, we provide:

- Figure 6 showing examples of the $f_c(x)$ function introduced in Lemma 6;
- In Section 7, theoretical results on the Hessian leading to proofs of Theorem 8 and (a stronger version of) Theorem 9 from §4.1, and Lemma 11 from §6; and
- In Section 8, additional illustrative experimental results with details on the Mpower selection heuristic.

Figure 6: Plots of upper bound $f_c(x)$ against $x$ for various values of $c$

## 7 The Hessian and Proofs of Earlier Results

In this Section, we first discuss properties of the Hessian in §7.1, then use these in §7.2 to prove Theorems 8 and 9, and Lemma 11. Define the *interior* to be all points $q \in (0,1)^n$. Recall that $r^*(x) = (r_1^*(q_i), \ldots, r_{i-1}^*(q_i), r_{i+1}^*(q_i), \ldots, r_n^*(q_i))$ with corresponding pairwise terms $\{\xi_{ij}^*\}$, is an arg max of $\mathcal{G}(q) = -\mathcal{F}(q)$ where $q_i$ is held fixed at a particular value. For notational convenience, define $r_i^* = q_i$.

### 7.1 Properties of the Hessian

From (Weller and Jebara, 2013), we have all terms of the Hessian matrix $H_{jk} = \frac{\partial^2 \mathcal{F}}{\partial q_j \partial q_k}$:

$$H_{jk} = \begin{cases} \frac{q_j q_k - \xi_{jk}}{T_{jk}} & \text{if } (j,k) \in \mathcal{E} \\ 0 & \text{if } (j,k) \notin \mathcal{E} \end{cases}, \quad H_{jj} = -\frac{d_j - 1}{q_j(1-q_j)} + \sum_{k \in \mathcal{N}(j)} \frac{q_k(1-q_k)}{T_{jk}}, \tag{5}$$

where $d_j = |\mathcal{N}(j)|$ is the degree of $j$, and $T_{jk} = q_j q_k (1-q_j)(1-q_k) - (\xi_{jk} - q_j q_k)^2 \geq 0$, with equality only at an edge (i.e. $q_j$ or $q_k \in \{0,1\}$). For an attractive edge $(j,k)$, in the interior, as shown in (Weller and Jebara, 2013, Lemma 14 in Supplement), $\xi_{jk} - q_j q_k > 0$ and hence $H_{jk} < 0$.

Now write

$$H_{jj} = \frac{1}{q_j(1-q_j)} + \sum_{k \in \mathcal{N}(j)} \left( \frac{q_k(1-q_k)}{T_{jk}} - \frac{1}{q_j(1-q_j)} \right). \tag{6}$$

Consider the term in large parentheses for some $k \in \mathcal{N}(j)$. First observe that the term is $\geq 0$, strictly $> 0$ in the interior, whether the edge is attractive or repulsive. Since $H_{jj} > 0$, on the surface $\frac{\partial \mathcal{F}}{\partial q_j}\Big|_{r^*} = 0$, we have

$$\frac{\partial r_j^*}{\partial r_k^*} = -\frac{H_{jk}}{H_{jj}}\Big|_{r^*}, \tag{7}$$

which also holds for $k = i$ where we define $r_i^* = q_i$.

Further, we may incorporate the term for $k$ to obtain

$$H_{jj} \geq \frac{1}{q_j(1 - q_j)} + \frac{q_k(1 - q_k)}{T_{jk}} - \frac{1}{q_j(1 - q_j)} = \frac{q_k(1 - q_k)}{T_{jk}},$$

with equality iff $j$ has no neighbor other than $k$ (again allowing $k = i$), in which case,

$$\frac{\partial r_j^*}{\partial r_k^*} = \frac{\xi_{jk}^* - r_j^* r_k^*}{r_k^*(1 - r_k^*)}. \tag{8}$$

We also show the following results, though the remainder of this Section §7.1 is not used until later when we prove Theorem 9 in §7.2.1.

Considering the term in large parentheses from (6), using the definition of $T_{jk}$, we may write

$$\left( \frac{q_k(1 - q_k)}{T_{jk}} - \frac{1}{q_j(1 - q_j)} \right) = \left( \frac{\xi_{jk} - q_j q_k}{T_{jk}} \right) \left( \frac{\xi_{jk} - q_j q_k}{q_j(1 - q_j)} \right) = -H_{jk}\beta_{j \to k}, \tag{9}$$

where we define $\beta_{j \to k} = \frac{\xi_{jk} - q_j q_k}{q_j(1 - q_j)}$, which as mentioned in the main paper after Theorem 8, is equal to $\frac{\mathrm{Cov}_q(X_j, X_k)}{\mathrm{Var}_q(X_j)}$, called in finance the beta of $X_k$ with respect to $X_j$. This is clearly positive for an attractive edge. We next show that the range of $\beta_{j \to k}$ is bounded, as would be expected for beta.

**Lemma 12.** *In the interior, for an edge* $(j, k)$*: if attractive,* $0 < \beta_{j \to k} \leq \frac{\alpha_{jk}}{\alpha_{jk} + 1} = 1 - e^{-W_{jk}} < 1$*; if repulsive,* $-1 < e^{W_{jk}} - 1 = \alpha_{jk} \leq \beta_{j \to k} < 0$*. In either case,* $|\beta_{j \to k}| = \left| \frac{\xi_{jk} - q_j q_k}{q_j(1 - q_j)} \right| \leq 1 - e^{-|W_{jk}|} < 1$.

*Proof.* This follows from (Weller and Jebara, 2013, Lemma 6) and the corresponding flipped result (Weller and Jebara, 2014, Lemma 10 in Supplement; consider each of the 2 cases for $p_{jk}$ therein). $\qquad\square$

Define $\beta_{j \to k}^* = \beta_{j \to k}\big|_{r^*}$. Regarding (8), note that $\beta_{j \to k}^* \geq \frac{\partial r_k^*}{\partial r_j^*}$ with equality iff $\mathcal{N}(k) = \{j\}$. This notation will become clear when we use it in §7.2.1 to prove Theorem 9.

## 7.2 Derivation of earlier results

Using the results of §7.1, we first provide a general Theorem from which Lemma 11 follows as an immediate corollary.

**Theorem 13.** *For any binary pairwise MRF where the Bethe free energy is convex, adding further variables to the model and holding them at fixed singleton marginal values (optimum pairwise marginals are computed using the formula of Welling and Teh, 2001), leaves the Bethe free energy over the original variables convex.*

*Proof.* The Bethe free energy is convex $\Leftrightarrow$ the Hessian is everywhere positive semi-definite. When new variables are added to the system, considering (5) and (6), the only effect on the sub-Hessian restricted to the original variables is potentially to increase the diagonal terms $H_{jj}$ for any original variable $j$ which is adjacent to a new variable. By Weyl's inequality, this can only increase the minimum eigenvalue of the sub-Hessian, and the result follows. $\qquad\square$

Since the Bethe free energy is convex for any model whose entire topology contains at most one cycle (Pakzad and Anantharam, 2002), Lemma 11 follows.

We next turn to Theorem 8, then use this to prove a stronger version of Theorem 9. Keep in mind that, as shown in (Weller and Jebara, 2013), each stationary point lies in an open region in the interior $q \in (0, 1)^n$. Further, as discussed in §4.1, we assume that at any $\arg\max$ point $r^*(q_i)$, the reduced Hessian $H_{\backslash i}$ is non-singular. Hence, writing $\nabla_{n-1}\mathcal{F}\big|_{q_i}$ for the $(n - 1)$-vector of partial derivatives $\frac{\partial \mathcal{F}(q)}{\partial q_j}\big|_{q_i}$ $\forall j \neq i$, there is an open region around any $(q_i, r^*(q_i))$ where the function $\nabla_{n-1}\mathcal{F}\big|_{q_i} = 0$ may be well approximated by an invertible linear function, allowing us to solve

(as in the implicit function theorem) for the total derivatives $\frac{dr_j^*}{dq_i}$ as the unique solutions to the linear system $\frac{dr_j^*}{dq_i} = \frac{\partial r_j^*}{\partial q_i} + \sum_{k \notin \{i,j\}} \frac{\partial r_j^*}{\partial r_k^*} \frac{dr_k^*}{dq_i} \; \forall j \neq i$, where here $\frac{\partial r_j^*}{\partial r_k^*}$ always means on the surface $\nabla_{n-1} \mathcal{F} \big|_{q_i} = 0$. In addition, since $H_{\backslash i}$ is real, symmetric, positive definite, with all main diagonal $\geq 0$ and all off-diagonal $\leq 0$, it is an M-matrix (indeed a Stieltjes matrix), which we shall use in §7.2.1. We assume these points for the rest of this Section.

**Notation:**    Let $D_j = \frac{dr_j^*}{dq_i}$, and $\partial_{jk} = \frac{\partial r_j^*}{\partial r_k^*}$, so $D_j = \sum_{k \notin \{i,j\}} \partial_{jk} D_k + \partial_{ji} \; \forall j \neq i$. For notational convenience, define $r_i^* = q_i$ and take $D_i = 1$. Let $[n] = \{1, \ldots, n\}$ and $[n] \setminus i = \{1, \ldots, n\} \setminus \{i\}$. Note that $\partial_{jk} = \frac{\partial r_j^*}{\partial r_k^*} \leq \beta_{k \to j}^*$ (equality iff $j$ has no neighbor other than $k$), as defined above. We shall write Hessian terms such as $H_{jk}$ to mean $H_{jk}\big|_{r^*}$ where this is implied by the context.

We first need the following Lemma.

**Lemma 14.** *Consider a MRF with $n$ variables, where then one more variable $X_{n+1}$ is added with singleton marginal $r_{n+1}^*$, adjacent to exactly one of the original $n$ variables, say $X_a$ with $a \in [n]$ (note we allow $a = i$), then: $D_1, \ldots, D_n$ are unaffected, and $D_{n+1} = \frac{\xi_{a,n+1}^* - r_a^* r_{n+1}^*}{r_a^*(1 - r_a^*)} D_a$.*

*Proof.* We have the linear system $D_j = \sum_{k \notin \{i,j\}} \partial_{jk} D_k + \partial_{ji} \; \forall j \in [n] \setminus i$. When $X_{n+1}$ is added, this yields a new equation for $D_{n+1}$, which as shown in (8), is $D_{n+1} = \frac{\xi_{a,n+1}^* - r_a^* r_{n+1}^*}{r_a^*(1 - r_a^*)} D_a$, and the only other equation that changes is the one for $D_a$, where we write $\partial_{ak}'$ and $\partial_{ai}'$ for the new coefficients. Hence, it is sufficient to show that the earlier solutions for $D_1, \ldots, D_n$ satisfy the new equation for $D_a$, i.e. if $D_a = \sum_{k \in [n+1] \setminus \{i,a\}} \partial_{ak}' D_k + \partial_{ai}'$.

Observe from (7) that $\partial_{ak}' = \partial_{ak} H_{aa} / H_{aa}' \; \forall k \in [n]$, where $H_{aa}'$ incorporates the new $X_{n+1}$ variable. Hence,

$$
\begin{aligned}
\sum_{k \in [n+1] \setminus \{i,a\}} \partial_{ak}' D_k + \partial_{ai}' &= \frac{H_{aa}}{H_{aa}'} \left( \sum_{k \notin \{i,j\}} \partial_{ak} D_k + \partial_{ai} \right) + \partial_{a,n+1}' D_{n+1} \\
&= \frac{H_{aa}}{H_{aa}'} D_a + \frac{\xi_{a,n+1}^* - r_a^* r_{n+1}^*}{T_{a,n+1} H_{aa}'} \frac{\xi_{a,n+1}^* - r_a^* r_{n+1}^*}{r_a^*(1 - r_a^*)} D_a \quad \text{by (7), (5) and just above} \\
&= \frac{D_a}{H_{aa}'} \left[ H_{aa} + \frac{(\xi_{a,n+1}^* - r_a^* r_{n+1}^*)^2}{T_{a,n+1} r_a^*(1 - r_a^*)} \right] \\
&= \frac{D_a}{H_{aa}'} \left[ H_{aa} + \left( \frac{r_{n+1}^*(1 - r_{n+1}^*)}{T_{a,n+1}} - \frac{1}{r_a^*(1 - r_a^*)} \right) \right] \quad \text{(definition of } T_{a,n+1}) \\
&= \frac{D_a}{H_{aa}'} \left[ H_{aa} + (H_{aa}' - H_{aa}) \right] = D_a \qquad \qquad \square
\end{aligned}
$$

Theorem 8 may now be proved by induction on $|C_k|$. The base case $|C_k| = 1$ follows from (8). The inductive step follows from Lemma 14 by considering a leaf.

### 7.2.1   Proof of (stronger version of) Theorem 9:

As above, we have the linear system given by the following equations:

$$
D_j = \sum_{k \notin \{i,j\}} \partial_{jk} D_k + \partial_{ji} \quad \forall j \neq i \qquad\qquad \Leftrightarrow -\partial_{ji} = \sum_{k \neq i} [\partial_{jk} - \delta_{jk}] D_k \qquad (10)
$$

with $\partial_{jk} = \frac{\partial r_j^*}{\partial r_{k*}} = -\frac{H_{jk}}{H_{jj}} \; k \notin \{i,j\}, \; \partial_{jj} := 0, \qquad \partial_{ji} = \frac{\partial r_j^*}{\partial q_i} = -\frac{H_{ji}}{H_{jj}}, \; \delta_{jk} = \begin{cases} 1 & j = k \\ 0 & j \neq k \end{cases}$.

Hence we may rewrite (10), multiplying by $-H_{jj}$, to give the equivalent system

$$
\sum_{k \neq i} H_{jk} D_k = -H_{ji} \quad \forall j \neq i \qquad\qquad (11)
$$

Note equation (11) makes intuitive sense: for each variable $X_j$, we have $\mathcal{F}_j = 0$ at a stationary point, then taking the total derivative with respect to $q_i$ gives $H_{ji} + \sum_{k \neq i} H_{jk} D_k = 0$.

By Theorem 8, we have the complete solution vector $D_k$ $\forall k \neq i$ provided the topology is acyclic. In this setting, we rewrite the result of Theorem 8 using the $\beta^*$ notation from above: $D_k = \prod_{(s \to t) \in P(i \rightsquigarrow k)} \beta^*_{s \to t}$, where here $P(i \rightsquigarrow k)$ is the *unique* path from $i$ to $k$.

For a general graph, there may be many paths from $i$ to $k$. Let $\Pi(i \rightsquigarrow k)$ be the set of all such directed paths. For any $r^*$, for any particular path $P(i \rightsquigarrow k) \in \Pi(i \rightsquigarrow k)$, define its *weight* to be $W[P(i \rightsquigarrow k)] = \prod_{(s \to t) \in P(i \rightsquigarrow k)} \beta^*_{s \to t}$. We shall prove the following result:

$$D_k \geq \max_{P(i \rightsquigarrow k) \in \Pi(i \rightsquigarrow k)} W[P(i \rightsquigarrow k)]. \tag{12}$$

Note this is clearly stronger than Theorem 9 since $\forall j \in \mathcal{N}(i)$, the path going directly $i \to j$ is one member of $\Pi(i \rightsquigarrow j)$, though in general there may be many others.

For any particular $r^*$, let $G'$ be the weighted directed graph formed from the topology of the MRF by replacing each undirected edge $s - t$ by two directed edges: $s \to t$ with weight $\beta^*_{s \to t}$ and $t \to s$ with weight $\beta^*_{t \to s}$. Note that in an attractive model, all $\beta^*_{s \to t} \in (0, 1)$, see Lemma 12.

It is a simple application of Dijkstra's algorithm to construct from $G'$ a tree of all maximum weight directed paths from $i$ to each vertex $j \neq i$, which we call $\mathcal{T}$.[6] (For our purpose we just need to know that such a tree $\mathcal{T}$ exists.)

We want to solve (11), which we write as $H_{\backslash i} D = -H_i$, where we want to solve for $D$, which is the vector of $D_k$ $\forall k \neq i$, and $H_i$ is the $i$th column of $H$ without its $i$th element. Let $H^{\mathcal{T}}_{\backslash i}$ be the reduced Hessian for the model on $\mathcal{T}$ (which is missing some edges), and $H^{\mathcal{T}}_i$ be the $i$th column of the Hessian for the model on $\mathcal{T}$ without its $i$th element. In the sub-model with only the edges of $\mathcal{T}$, by construction and Theorem 8, $D^{\mathcal{T}}_k = \max_{P(i \rightsquigarrow k) \in \Pi(i \rightsquigarrow k)} W[P(i \rightsquigarrow k)]$. Hence, it is sufficient to show that adding the extra edges from $\mathcal{T}$ to $G$ cannot decrease any $D_k$. This forms the remainder of the proof, where we shall require the following nonsingular M-matrix property of $H_{\backslash i}$: its inverse is elementwise non-negative (Fan, 1958, Theorem 5').

Let $\Delta = H_{\backslash i} - H^{\mathcal{T}}_{\backslash i}$ (this accounts for edges in $E(G) \setminus E(\mathcal{T})$ not incident to $i$), $\eta = H_i - H^{\mathcal{T}}_i$ (this accounts for edges in $E(G) \setminus E(\mathcal{T})$ incident to $i$) and $\delta = D - D^{\mathcal{T}}$. We must show that $\delta \geq 0$ elementwise. We have $H^{\mathcal{T}}_{\backslash i} D^{\mathcal{T}} = -H^{\mathcal{T}}_i$ and $H_{\backslash i} D = -H_i$, hence $H^{\mathcal{T}}_{\backslash i} D^{\mathcal{T}} - \eta = -H^{\mathcal{T}}_i - \eta = -H_i = H_{\backslash i} D = (H^{\mathcal{T}}_{\backslash i} + \Delta)(D^{\mathcal{T}} + \delta)$, hence $-\eta = (H^{\mathcal{T}}_{\backslash i} + \Delta)\delta + \Delta D^{\mathcal{T}} \Leftrightarrow \delta = (H_{\backslash i})^{-1}(-\eta - \Delta D^{\mathcal{T}})$. Thus, it is sufficient to show that the $(n-1)$ vector $-\eta - \Delta D^{\mathcal{T}}$ is elementwise non-negative.

Recall (5) and (9). $-\eta - \Delta D^{\mathcal{T}}$ may be written as the sum of $-\eta_e - \Delta_e D^{\mathcal{T}}$, with one $\eta_e$ and $\Delta_e$ for each edge $e = (s, t)$ in $E(G) \setminus E(\mathcal{T})$. For each such edge $e$, we have 2 cases:

*Case 1, $i \notin \{s, t\}$*: $\eta_e = 0$; $\Delta_e$ has only 4 non-zero elements, at locations $(s, s), (s, t), (t, s), (t, t)$. Showing only these elements,

$$\Delta_e = \begin{matrix} s \\ t \end{matrix} \begin{pmatrix} \overset{s}{-H_{st}\beta^*_{s \to t}} & \overset{t}{H_{st}} \\ H_{st} & -H_{st}\beta^*_{t \to s} \end{pmatrix} = -H_{st} \begin{matrix} s \\ t \end{matrix} \begin{pmatrix} \overset{s}{\beta^*_{s \to t}} & \overset{t}{-1} \\ -1 & \beta^*_{t \to s} \end{pmatrix}, \text{ where } -H_{st} > 0 \text{ for an attractive edge.}$$

Hence, $-\eta_e - \Delta_e D^{\mathcal{T}}$ is 0 everwhere except element $s$ which is $-H_{st}(D^{\mathcal{T}}_t - D^{\mathcal{T}}_s \beta^*_{s \to t})$, and element $t$ which is $-H_{st}(D^{\mathcal{T}}_s - D^{\mathcal{T}}_t \beta^*_{t \to s})$. Observe that both expressions are $\geq 0$ by construction of $\mathcal{T}$ (for example, considering the first bracketed term, observe that $D^{\mathcal{T}}_t$ is the maximum weight of a path from $i$ to $t$, whereas $D^{\mathcal{T}}_s \beta^*_{s \to t}$ is the weight of a path to $t$ going through $s$).

*Case 2, $i \in \{s, t\}$*: WLOG suppose the edge is $(i, s)$. $-\eta_e$ is zero everywhere except element $s$ which is $-H_{is}$ (positive). $\Delta_e$ has just one non-zero element at $(s, s)$ which is $-H_{is}\beta^*_{s \to i}$. Hence, $-\eta_e - \Delta_e D^{\mathcal{T}}$ is 0 everwhere except element $s$ which is $-H_{is}(1 - D^{\mathcal{T}}_s \beta^*_{s \to i}) > 0$ by Lemma 12.

This completes the proof.

(a) attractive $\log Z$, $T_{max} = 2$    (b) attractive margs, $T_{max} = 2$    (c) general $\log Z$, $T_{max} = 0.1$    (d) general margs, $T_{max} = 0.1$

Figure 7: Average errors vs true, **complete graph on $n = 10$**. Consistent legend throughout.

(a) attractive $\log Z$, $T_{max} = 2$    (b) attractive margs, $T_{max} = 2$    (c) general $\log Z$, $T_{max} = 0.1$    (d) general margs, $T_{max} = 0.1$

Figure 8: Average errors vs true, **random graph on $n = 10$, $p = 0.5$**. Consistent legend throughout.

(a) attractive $\log Z$, $T_{max} = 2$    (b) attractive margs, $T_{max} = 2$    (c) general $\log Z$, $T_{max} = 0.1$    (d) general margs, $T_{max} = 0.1$

Figure 9: Average errors vs true, **random graph on $n = 50$, $p = 0.1$**. Consistent legend throughout.

Figure 10: 'Lamp' topology.
maxW is likely to choose $x_6$ since it has the highest degree, but $x_4$ is typically a better choice since it lies on cycles. Mpower can recognize this and make a better choice.

## 8   Additional Experiments

All of the experiments reported in §5 were also run at other settings. In particular, the earlier results show the poor performance of the standard Bethe approximation in estimating singleton marginals for attractive models with low singleton potentials, and indicate how clamping repairs this. Here, in Figures 7-9, we show results for the same topologies using the higher singleton potentials $T_{max} = 2$ for attractive models, and also show results with low singleton potentials $T_{max} = 0.1$ for general (non-attractive) models.

Note that in some examples of attractive models, when the 'worst clamp' variable was clamped, the resulting Bethe approximation to $\log Z$ appears to worsen (see Figure 9a), which seems to conflict with Theorem 5. The explanation is that in these examples, Frank-Wolfe is failing to find the global Bethe optimum, as was confirmed by spot checking.

Next we show results for a particular fixed topology we call a 'lamp', see Figure 10, which illustrates how maxW can sometimes select a poor variable to clamp. We explain the Mpower selection heuristic and demonstrate that it performs much better on this topology.

(a) attractive log $Z$, $T_{max} = 2$  (b) attractive margs, $T_{max} = 2$  (c) general log $Z$, $T_{max} = 2$  (d) general margs, $T_{max} = 2$

Figure 11: Average errors vs true, **'lamp' topology $T_{max} = 2$**. Consistent legend throughout. Mpower performs well, significantly better than maxW.

(a) attractive log $Z$, $T_{max} = 0.1$  (b) attractive margs, $T_{max} = 0.1$  (c) general log $Z$, $T_{max} = 0.1$  (d) general margs, $T_{max} = 0.1$

Figure 12: Average errors vs true, **'lamp' topology $T_{max} = 0.1$**. Consistent legend throughout. Mpower performs well, significantly better than maxW for $\log Z$.

## 8.1 Mpower heuristic

We would like an efficient way to select a variable to clamp which lies on many heavy simple cycles. One problem is how to define heavy. Even with a good definition, it is still NP-hard to search over all simple cycles. The idea for Mpower is as follows: assign each edge $(i, j)$ a weight based on $|W_{ij}|$ and create a matrix $M$ of these weights. If $M$ is raised to the $k$th power, then the $i$th diagonal element in $M^k$ is the sum over all paths of length $k$ from $i$ to $i$ of the product of the edge weights along the path. Ideally, we might consider the sum $\sum_{k=1}^{\infty} M^k$ and use the diagonal elements to rank the vertices, choosing the one with highest total score. Recalling (12), it is sensible to assign edge weights $M_{ij}$ based on possible $\beta_{i \to j}^*$ values. Given Lemma 12, a first idea is to use $1 - e^{-|W_{ij}|}$.

However, we'd like to be sure that the matrix series $\sum_{k=1}^{\infty} M^k$ is convergent, allowing it to be computed as $(I - M)^{-1} - I$ (since we shall be interested only in ranking the diagonal terms, in fact there is no need to subtract $I$ at the end). Thus, we need the spectral radius $\rho(M) < 1$. A sufficient condition is that all row sums are $< 1$. Since each term $1 - e^{-|W_{ij}|} < 1$ and there at most $n - 1$ such elements in any row, our first heuristic was to set $M_{ij} = \frac{1}{n-1}(1 - e^{-|W_{ij}|})$. We then made two adjustments.

First, note that the series $\sum_{k=1}^{\infty} M^k$ overcounts all cycles, though at an exponentially decaying rate. It is hard to repair this. However, it also includes relatively high value terms coming from paths from $i$ to any neighbor $j$ and straight back again, along with all powers of these. We should like to discard all of these, hence from each $i$th diagonal term of $(I - M)^{-1}$, we subtract $s_i/(1 - s_i)$, where $s_i$ is the $i$th diagonal term of $M^2$. This is very similar to the final version we used, and gives only very marginally worse results on the examples we considered.

For our final version, we observe that $1 - e^{-|W_{ij}|}$ decays rapidly, and $\approx \tanh \frac{|W_{ij}|}{2}$. Given the form of the loop series expansion for a single cycle, which contains $\tanh \frac{W_{ij}}{4}$ terms (Weller et al., 2014, Lemma 5), we tried instead using $M_{ij} = \frac{1}{n-1} \tanh \frac{|W_{ij}|}{4}$, and it is for this heuristic that results are shown in Figures 11 (for $T_{max} = 2$) and 12 (for $T_{max} = 0.1$). Observe that for this topology, Mpower performs close to optimally (almost the same results as for best Clamp), significantly outperforming maxW in most settings. Note, however, that in the experiments on random graphs reported in §5, Mpower did not outperform the simpler maxW heuristic. In future work, we hope to improve the selection methods.

## Footnotes

[6]We want the max of the prod of edge weights $\Leftrightarrow$ max of the log of the prod of edge weights $\Leftrightarrow$ max of the sum of the log of edge weights (all negative) $\Leftrightarrow$ min of the sum of - log of the edge weights (all positive); so really we construct the usual shortest directed paths tree using - log of the edge weights, which are all positive.