[Reviews · NeurIPS 2014]

Submitted by Assigned_Reviewer_31

The paper introduces a series of bounds on some approximations of the partition function for graphical models. They first bound from below the lower bound provided by 'ExactOnTree' approximations, and then show that, for a binary pariwise models, clamping (i.e. fixing the value of) a node improves the quality of such a lower bound.
The paper is clear and interesting. Insight about the quality of lower bounds that are used in many applications is obiously usefull. The bounds provided here are new, at least to me. The proofs are also clear. The author also provide some intuitive explanation about when clamping can improve or degradate the approximation.
My main concern is about the Experiment section, which I found unclear and hard to read, probably due to lack of space.
Summary: This paper is the best of the four articles I had to review.

Submitted by Assigned_Reviewer_33

The paper presents a novel proof to the recent result by Ruozzi: The Bethe free energy fixed points are lower bounds on the partition function for attractive models. The basic idea is that clamping variables only increases the Bethe partition function. The proof is then completed when enough variables are clamped.

The paper is clearly written and the idea of using clamping to the Bethe entropy is original, surprising and very interesting. I might suggest the authors not to use Z in Theorem 4 unless it always refers to the partition function.
Summary: The paper is original and the proof is intuitive. It is an original direction and the power of clamping is yet to be fully understood.

Submitted by Assigned_Reviewer_37

Review for On the Power of Clamping

Summary: This paper presents some deep technical results on inference
in a particular class of undirected graphical models. By clamping
well-chosen variables, a complex graph can be split into smaller
graphs, which can each be processed independently, and the results
combined. The authors present approximation bounds for the partition
function in such an approach, and demonstrate that clamping improves
approximation error.

This paper is a well-written deep dive into a specific inference
approach on a specific model class (binary pairwise model with
attractive potentials). Proofs of the main technical results are the
centerpiece of the paper, although some simple experiments demonstrate
that the approach can have practical value by reducing approximation
error.

Overall I found this paper well-done, although a bit dense at times.
While it's unclear to me how generally interesting these results are
(given the fairly limited class of applicable problems), the material
pushes the state-of-the-art in a long line of similar work that the
community considers valuable.

One weakness of this paper is the general lack of connection to other
approximation methods. As the authors note, the idea of conditioning
on a variable (or set of variables) and approximating over the rest
has a long and fruitful history. Given that rich and varied
literature, it would have been interesting to see how other methods
compare to clamping+Bethe, either theoretically or practically. The
authors do compare to TRW in the experiments, but I'm left wondering
if there are other serious approximation methods for the same class of
models, and if so, what their performance/guarantees might look like.

Along those lines, I strongly encourage the authors to pursue the idea
outlined in the last paragraph: "It is tempting to speculate that
similar results may be found for other approximations. For example,
for some methods that upper bound the partition function, such as TRW,
it may be possible to show that a lower (hence better) approximation
is always achieved when a variable is clamped." That would be
interesting indeed.

Minor comments:

You prove that Z_B <= Z, and that Z_B^i >= Z_B. Perhaps I missed it,
but can you say anything about the relationship between Z_B^i and Z?
Is clamping upper-bounded by Z?
Summary: The authors provide some approximation bounds on clamping+Bethe approximate inference in pairwise binary models with attractive potentials. It is generally well-done, although somewhat narrow in scope.
Author Feedback
Author rebuttal: Overall
+Many thanks to all reviewers for your time and comments.
+Please note that, to our knowledge, several of the techniques used are novel and interesting in their own right. For example, we believe it is original to combine the clamping idea with its impact on the variational inference problem, and examine the effect on the location of an optimum and the entropy of a single variable, as that variable's singleton marginal is varied. Also examining the properties of the Hessian of the free energy approximation yields powerful results such as Lemma 11 on the convexity of the Bethe free energy over any sub-model with one cycle, while other variables are held to fixed singleton marginals, which may be helpful for optimizing the Bethe free energy.
+We believe our approach, by providing new tools and perspective on the problem, will directly lead to better algorithms for selecting which variable to clamp in future work. We proposed the maxW and Mpower approaches and demonstrated that they are useful. We shall describe them more fully in the Appendix and further show that Mpower can outperform maxW significantly on some topologies. Mpower correctly selects nodes on heavy cycles, whereas the maxW approach favors nodes with many edges and cannot check if those nodes lie on a cycle or not. Mpower is still fairly quick to compute (it requires one matrix inversion).
+Given the widespread application of belief propagation, we believe our results are important, and in addition, the new techniques and perspectives may prove valuable across other areas.

Reviewer 31
+Thank you for your comments. We shall make the experiment results clearer, including using larger fonts, and add more detail in the Appendix.

Reviewer 33
+Thank you for your comments. We shall be sure to use Z unambiguously in the camera-ready.

Reviewer 37
+Thank you for your comments. Please note that binary pairwise models are 'positive universal', i.e. any discrete model with strictly positive probabilities can be reduced to one, and 'almost universal' in that even if a general model has 0 probabilities, it is possible via Leisink's construction to find a binary pairwise model (which might be significantly larger) that approximately reduces it to arbitrary precision (Eaton and Ghahramani 2013). Also our first result, demonstrating a lower bound for any ExactOnTrees approximate partition function in terms of a fraction of the true partition function, applies to general models (not necessarily attractive) with any number of variables.
+On comparing against other approximation methods, there is a significant literature on this, see e.g. Sudderth et al. NIPS 07, Sontag and Jaakkola NIPS 07, Meshi et al. UAI 2009, and Weller et al. UAI 2014, where a general conclusion is that Bethe performs remarkably well, typically beating other approaches. Hence the ability to significantly improve on Bethe by simply clamping a few variables, as demonstrated in our paper, is of great interest.
+Response to Minor comment:
Theorem 5 proves that Z_B \leq Z_B^i for any variable X_i, where Z_B^i is the sum of optimum Bethe partition functions over each of the two sub-models formed by clamping X_i=0 and X_i=1 respectively.
The idea behind the proof of Theorem 10, which proves that Z_B \leq Z, is to lever Theorem 5 by iterating the result, clamping additional variables until we reach a situation where, after those clamped variables have been removed from the model topology, the remaining graph is a tree, on which Bethe is exact because it is ExactOnTrees.
So, we have Z_B \leq Z_B^i \leq Z_B^{ij} \leq Z_B^{ijk} [where at each stage, we double the number of sub-partition functions that must be summed, each one of which is over a sub-model with 1 more variable clamped] ... \leq Z
In particular, indeed yes, Z_B^i \leq Z.